# Perceive Anything: Recognize, Explain, Caption, and Segment Anything in Images and Videos

**Weifeng Lin[1]\*** **Xinyu Wei[3]\*** **Ruichuan An[4]\*** **Tianhe Ren[2]\*** **Tingwei Chen[1]**
**Renrui Zhang[1]** **Ziyu Guo[1]** **Wentao Zhang[4]** **Lei Zhang[3]** **Hongsheng Li[1]†**

[1]CUHK  [2]HKU  [3]PolyU  [4]Peking University

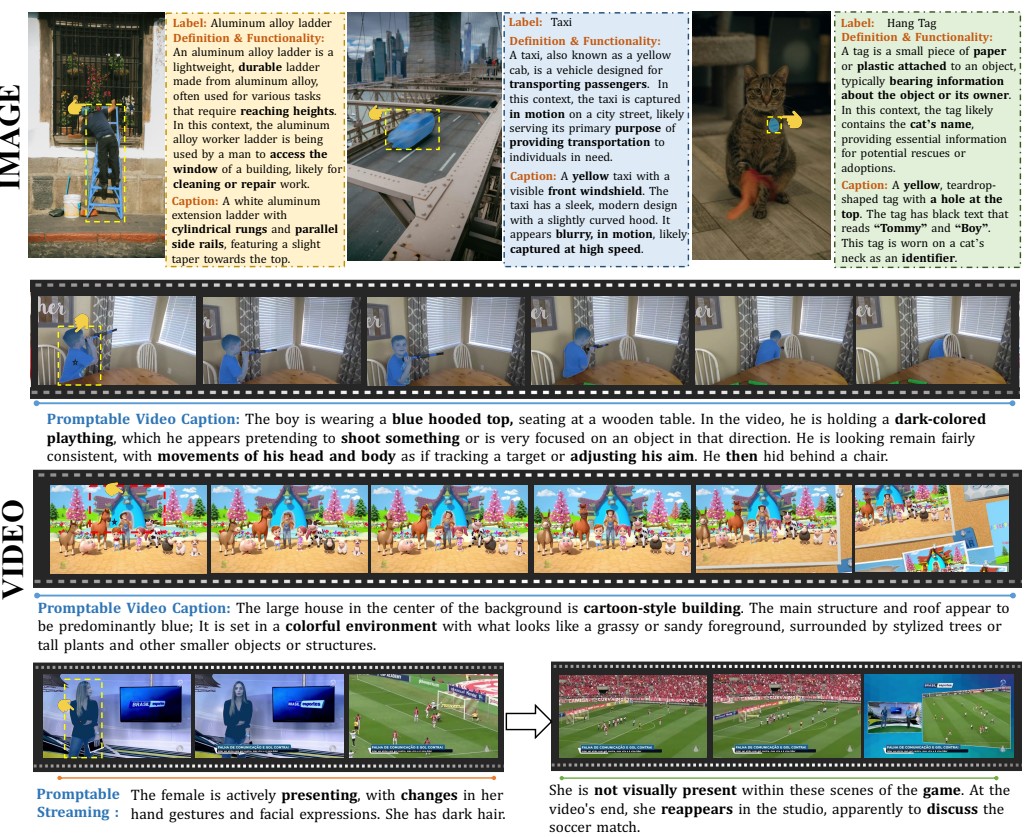

Figure 1: **Perceive Anything Model (PAM):** PAM accepts various visual prompts (such as clicks, boxes, and masks) to produce region-specific information for images and videos, including masks, category, label definition, contextual function, and detailed captions. The model also handles demanding region-level streaming video captioning.

## Abstract

We present Perceive Anything Model (PAM), a conceptually straightforward and efficient framework for comprehensive region-level visual understanding in images and videos. Our approach extends the powerful segmentation model SAM 2 by integrating Large Language Models (LLMs), enabling simultaneous object

---

\*Core Contributor
†Corresponding Authors

39th Conference on Neural Information Processing Systems (NeurIPS 2025).

segmentation with the generation of diverse, region-specific semantic outputs, including categories, label definition, functional explanations, and detailed captions. A key component, Semantic Perceiver, is introduced to efficiently transform SAM 2's rich visual features, which inherently carry general vision, localization, and semantic priors into multi-modal tokens for LLM comprehension. To support robust multi-granularity understanding, we also develop a dedicated data refinement and augmentation pipeline, yielding a high-quality dataset of 1.5M image and 0.6M video region-semantic annotations, including novel region-level streaming video caption data. PAM is designed for lightweightness and efficiency, while also demonstrates strong performance across a diverse range of region understanding tasks. It runs $1.2-2.4\times$ faster and consumes less GPU memory than prior approaches, offering a practical solution for real-world applications. We believe that our effective approach will serve as a strong baseline for future research in region-level visual understanding. Code, model and data are available at: https://Perceive-Anything.github.io

# 1 Introduction

The vision community has rapidly witnessed advances in vision foundation models, such as SAM [34] and SAM 2 [52], which have dramatically improved interactive object segmentation performance in images and videos. These models offer remarkable precision in localizing arbitrary objects based on various visual prompts. However, they typically lack deep semantic understanding of the segmented regions, elucidating what these regions mean or how they function in context remains a challenging problem.

Recent studies seek to endow Vision–Language Models (VLMs) with region-level understanding capability through visual prompts. As illustrated in Fig. 2, current methods can be grouped into three paradigms: *(1)* textual encoding [63, 78, 86, 44], which encode 2-D bounding-box coordinates as natural-language strings inside the prompt, thereby supplying no explicit region prior; *(2)* visual-prompt encoding (VPE) [41, 51], which introduce extra module to embed regional image features and positional features; *(3)* RoI/segmentation-based encoding [38, 77, 83, 80, 29], which utilize an external mask generator to concatenate image embedding and mask embedding. While these methods show promise, they often present several limitations: *(i)* they usually generate only limited semantic outputs—often just category labels or short captions [26, 88, 69, 67]; *(ii)* their designs are modality-specificl, focusing on one single visual modality (image or video), offering limited generality [63, 78, 77, 80, 81]. *(iii)* they rely on external segmentation models to supply masks, a serial design that adds computational overhead and makes overall performance sensitive to mask quality [80, 81, 38].

To address these challenges, we introduce the Perceive Anything Model (PAM), an end-to-end region-level vision-language model designed for fast and comprehensive fine-grained visual understanding across both images and videos, encompassing capabilities such as predicting categories, explaining the definition and contextual function of identified regional elements, and generating detailed descriptions of specific regions. Rather than redesigning model architecture from scratch, our approach efficiently extends the SAM 2 framework with Large Language Models (LLMs) to support semantic understanding. Specifically, we introduce a Semantic Perceiver that acts as a essential bridge, effectively leveraging rich intermediate visual features from the SAM 2 backbone to integrate general vision, localization, and semantic priors into visual tokens. These tokens are subsequently processed by the LLM to generate a diverse semantic outputs. Furthermore, PAM features a parallel design for its mask and semantic decoders, enabling simultaneous generation of region masks and semantic content, thereby improving computational efficiency.

To ensure PAM's robustness in understanding region-level multi-dimensional semantic granularity, high-quality training data is an essential component. While multiple existing datasets [6, 32, 36, 43, 29, 68] provide region-semantics annotations, we noticed that they are often overly coarse, limiting their utility for fine-grained understanding tasks. Therefore, to construct high-quality training data, we develop an advanced data refinement and augmentation pipeline that leverages leading VLMs (e.g., GPT-4o [27]) and human expert validation to refine and augment existing region-level annotated datasets. For images, we generate annotations at multiple distinct semantic granularities for each specific region: a fine-grained category label, a context-aware definition that clarifies the region's role

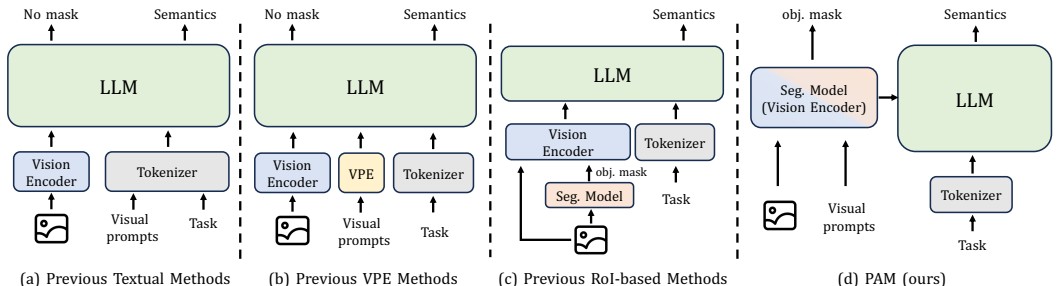

Figure 2: **Previous Paradigms vs. Our Paradigm (PAM).** (a & b) Textual/VPE methods provide region understanding using positional embeddings but typically lack simultaneous object masks. (c) RoI/Segmentation-based methods use external segmenter for object masks, subsequently fusing image and mask embeddings. (d) In contrast to previous paradigms, our method directly treats the Seg. model as vision encoder. It effectively leverages the rich visual embeddings from the robust segmentation model and features a parallel design for its mask and semantic decoders.

or function within the scene, and detailed descriptions. For videos, we refined original coarse-level annotations from referring video detection and segmentation dataset [64, 58, 18, 71, 17] into detailed, temporally-aware region-level captions. Furthermore, we pioneered the development of event-based, region-level streaming video caption data. To the best of our knowledge, this is the first work to construct such a dataset, enabling the model to support streaming video region captioning. Notably, we also generate bilingual (English and Chinese) versions of each data annotation to equip the model with multilingual response capabilities. This process yields a final high-quality dataset comprising 1.5M image-region-semantics triples and 0.6M video-region-semantics triples.

Our experimental results demonstrate that PAM delivers robust performance across a diverse range of regional understanding tasks for both images and videos, while operating $1.2-2.4\times$ faster and consuming less GPU memory compared to prior models. We believe our model, dataset, and insights will significantly advance research in this domain and broadly benefit the vision-language community.

## 2 Related Work

**Interactive Image and Video Object Segmentation.** Interactive object segmentation has progressed rapidly in recent years. Early methods—such as Graph Cut [5] and Active Contours [9]—relied on manual annotations (e.g., foreground/background clicks). Inspired by the paradigm of pre-training autoregressive Transformer architectures on large-scale data in language modeling, the Segment Anything Model (SAM) [34] revolutionized user–model interaction by ingesting multiple visual prompts and segmenting arbitrary objects in a class-agnostic manner. SAM 2 [52] extended this capability to video, enabling real-time processing of arbitrarily long sequences with strong generalization. Subsequent research [33, 82, 87, 54, 13, 74], while retaining the core architecture, has further improved the accuracy and efficiency of this model family.

**Region-level Vision-Language Models (VLMs).** Region-level understanding tasks, such as region classification and captioning, are fundamental in computer vision. Regarding image-based VLMs, recent researches [46, 12, 70, 84, 77, 80, 8, 41, 1] have demonstrated a notable trend towards enabling region-level understanding capabilities through spatial visual prompts. Furthermore, research has extended regional understanding to the video domain [78, 63, 81, 48], focusing on identifying and interpreting user-specified regions across temporal intervals. However, these approaches typically operate within a single modality (either image or video), and more complex tasks, such as region-level streaming video captioning—which requires continuously generating textual descriptions for specific regions as a video progresses—remain largely unaddressed.

**Streaming Video Captioning.** Streaming video captioning demands per-frame processing and rapid response times. Recent online video understanding models [16, 20] aim to identify the current action at each timestamp. Streaming video caption [89] propose to incorporate memory modules and develop specialized streaming decoding algorithms to support streaming captioning. VideoLLM online [11]

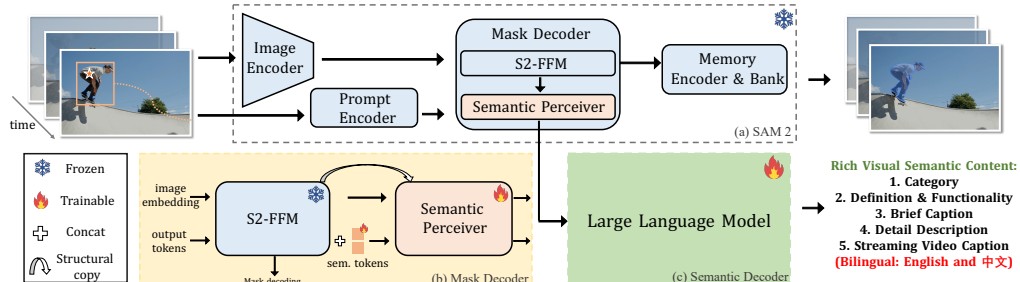

Figure 3: Overall Architecture of PAM.

further pioneered the use of LLMs to achieve free-form dialogue synchronized with the online video stream. However, these approaches predominantly focus on general event comprehension, leaving the continuous tracking and description of specific regions within a video stream as a significant unresolved challenge.

## 3  Perceive Anything Model (PAM)

Given visual prompts such as points, boxes, or masks to specify a region of interest, **Perceive Anything Model (PAM)** can simultaneously: *(1) Segment:* Generate precise segmentation masks for the indicated region within an image or throughout a video. *(2) Recognize:* Identify the category of the designated region or object. *(3) Explain:* Provide clear explanations of the region's or object's definition, attributes, and functionality within its given context. *(4) Caption:* Generate concise or detailed captions for the region within images, videos, and video streams.

### 3.1  Model Architecture

As illustrated in Fig. 3, our PAM can be divided into two parts. The first part is the SAM 2 framework, which comprises an image encoder, a prompt encoder, memory modules, and a mask decoder. This framework provides robust spatio-temporal visual feature extraction and segmentation capabilities. The second part is a semantic decoder, which is based on a large language model (LLM). Crucially, our proposed Semantic Perceiver acts as a bridge, effectively leverages intermediate visual features from the SAM 2 backbone and results in visual tokens. These tokens are subsequently processed by the LLM to generate diverse semantic outputs. For decoding, PAM features a parallel design for its mask and semantic decoders, enabling the simultaneous segmentation of objects while generating diverse semantic outputs of them. The design of components and training process are detailed below.

**Semantic Perceiver.**    As shown in Fig. 3(b) and Fig. 4, the architecture of Semantic Perceiver mirrors the SAM 2 Feature Fusing module (S2-FFM), employing a lightweight two-layer transformer with self-attention, cross-attention, and a point-wise MLP. Specifically, it receives two primary inputs: enhanced mask tokens from S2-FFM, which incorporate IoU and prompt tokens information and serve as unique identifiers for precise mask generation; and updated image embeddings after S2-FFM, capturing general visual context and implicit features enriched through interaction with mask tokens. Next, following [26, 28], we concatenate $N_s$ learnable semantic tokens with the enhanced mask tokens. Finally, through further attention mechanisms within the Semantic Perceiver, we can fetch visual tokens rich in both general visual and object-level localization information. Given an input of $N$ frames (where N=1 for a single image), Semantic Perceiver outputs two sets of 256-dimensional vectors: $64^2 \times N$ visual tokens and $N_s \times N$ semantic tokens ($N_s = 16$ by default).

**Projector.**    The projector preceding the LLM comprises two layers: a pixel shuffle operation and an MLP projector. For image inputs, we apply the pixel shuffle operation over adjacent $2 \times 2$ feature patches to downsample the number of visual tokens. For video inputs, the prompted frame is processed similarly with single image, while the remaining frames in the video clip undergo a more aggressive $4 \times 4$ pixel shuffle operation to significantly reduce visual tokens and further improve processing efficiency for semantic decoder. Subsequently, we use two distinct MLPs [45] to project visual and semantic tokens separately.

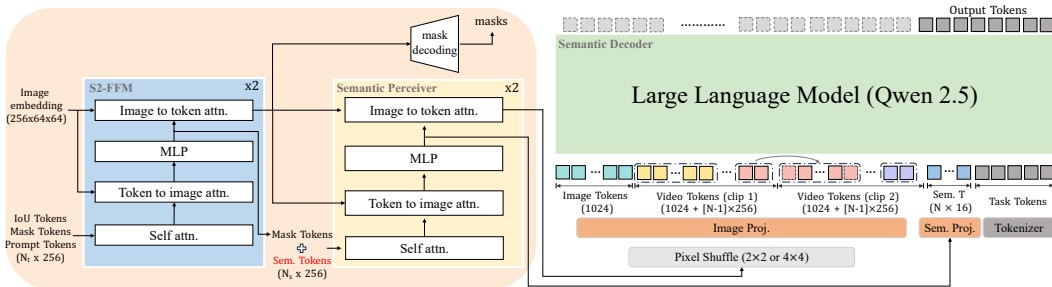

Figure 4: **Detailed illustration of our PAM workflow.** Semantic Perceiver first receives enhanced image embeddings and mask tokens from the S2-FFM and outputs enriched visual tokens and semantic tokens. These are subsequently fed into the semantic decoder for decoding.

**Semantic Decoder.** We adopt the pre-trained Qwen2.5 LLM [72] as our semantic decoder, leveraging its strong language processing capabilities. This decoder is responsible for interpreting the processed visual tokens and semantic tokens alongside task instructions to generate the desired semantic outputs.

**Streaming Video Encode and Decode.** Building upon the progressive introduction of historical information per frame via memory modules in SAM 2, we propose a straightforward strategy for region-level streaming video captioning without adding complex components. Specifically, an additional 2×2 pixel shuffle operation is applied to the last frame of each video clip. This leads to a greater density of visual tokens, improving the preservation of historical visual information. These tokens subsequently act as the initial frame for the next video clip and are processed by the LLM together with the remaining frames of that clip. This approach ensures that each clip is processed consistently and effectively passes crucial historical information from the previous clip into the next video clip. Additionally, we incorporate the previous textual description into the prompt to further augment contextual history, enhancing the model's comprehension and descriptive accuracy for ongoing events. In practice, our framework allows users to flexibly specify decode timestamps. Upon reaching a designated timestamp, the model describes the specified region within the temporal interval between the current timestamp and the previous one.

**Training Strategies.** We structure our training process using a three-stage curriculum learning approach, progressively enhancing the PAM's region-level visual understanding capabilities from images to video. In all training stage, the parameters of SAM 2 are frozen. The hyper-parameters for each training stage are summarized in Appendix A.

• *Stage 1: Image Pretraining and Alignment.* The initial training stage focuses on establishing robust alignment among visual tokens, semantic tokens and the language model's embedding space. The primary objective is to enable the model to effectively understand region-level image content. To this end, we utilize a large dataset of region-level image classification and captioning. During this stage, only the semantic perceiver and the projector are trained.

• *Stage 1.5: Video-Enhanced Pretraining and Alignment.* In this stage, we extend the initial image-based training by incorporating region-level video captions. This inclusion enables the model to comprehend dynamic scenes through the integration of spatio-temporal visual information. The trainable modules are the same as in Stage 1.

• *Stage 2: Multimodal Fine-Tuning.* The final stage employs supervised fine-tuning (SFT) to enable the model to perform diverse tasks and generate desired responses. This stage utilizes a high-quality dataset, which has been refined and augmented via our pipeline (Sec. 4). Training in this phase jointly involves the semantic perceiver, the projector, and the semantic decoder.

## 4 Data

To enhance PAM's comprehensive visual perception capabilities, we develop a robust data refinement and augmentation pipeline to curate a high-quality training dataset. This dataset is distinguished by three key features: *(1) Broad-ranging Semantic Granularities.* It provides diverse visual semantic

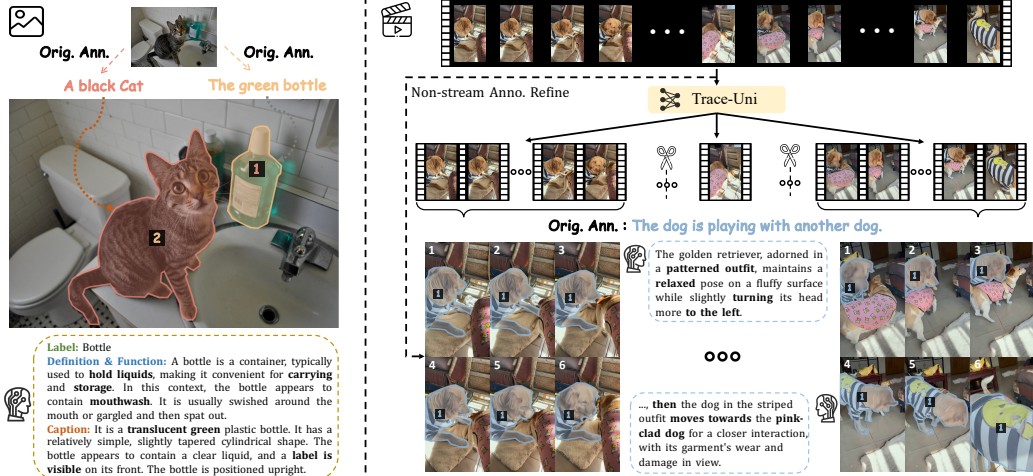

Figure 5: **Illustrative examples of our dataset construction pipeline.** The left panel displays image annotations; the right panel details annotations for non-streaming and streaming video.

annotations spanning from coarse-level (categories, definitions, contextual functionalities) to fine-grained (detailed descriptions) (Sec. 4.1). *(2) Regional Streaming Caption Annotations.* The first dataset to curate annotations specifically for streaming video region captioning (Sec. 4.2). *(3) Bilingual Annotations,* supporting both English and Chinese (App. B.2). The pipeline is detailed below, and additional information are available in Appendix B.

## 4.1 Image Dataset

**Regional Recognition, Explanation, and Caption.** For regional recognition, we utilize multiple instance detection and segmentation datasets [55, 35, 40, 23, 50, 66], along with scene text recognition datasets [56, 31, 30, 19, 24, 14, 76, 57, 4]. In this context, the bounding box or mask serves as the visual prompt input, and the label is treated as the output.

To achieve deep, fine-grained visual understanding beyond simple classification, we propose an enhanced pipeline that generates: clear conceptual explanations, contextual functional roles, and detailed descriptions for each specific region. This multi-dimensional information aims to significantly improve user comprehension, particularly for uncommon terms or unfamiliar subjects. To implement this, we utilize the latest VLMs for their extensive world knowledge and powerful visual understanding capabilities to assist refinement. Specifically, we apply the Set of Mask (SoM) method [75] to identify regions of interest, and use original annotations as context to guide models to produce desired responses, which then undergo manual quality assurance. An illustrative example is presented in Fig. 5(left). We present more details in Appendix B.1.

## 4.2 Video Dataset

**Region-level Video Caption.** To extend the model's regional captioning capabilities to video, we collected and analyzed several existing video datasets, including referring detection and segmentation datasets [71, 47, 18, 62, 58, 17, 85, 64], as well as the recent Sa2VA [79] annotations for the SA-V [53] dataset. These datasets, designed for detecting, segmenting, and captioning specific objects in videos based on textual descriptions, often contain descriptions that are overly coarse, simplistic, inaccurate, or predominantly static, neglecting essential temporal details such as object motion, interactions, and state changes throughout the video.

To address the existing limitations, we propose the **storyboard-driven caption expansion** method. This process involves several key stages: *(1) Keyframe Sampling*: Six keyframes are uniformly extracted from each video. *(2) Storyboard Synthesis*: These extracted keyframes are combined to form a high-resolution composite image, presented in a storyboard format (as illustrated in Fig. 5). *(3) Object-Centric Highlighting*: Within this composite image, each individual frame specifically highlights the target object using a colored bounding box or mask, implemented by

| Model | Classification | | | | OCR | |
| | LVIS | | PACO | | COCO Text | Total-Text |
| | Semantic Sim. | Semantic IoU | Semantic Sim. | Semantic IoU | Acc.(%) | Acc.(%) |
|---|---|---|---|---|---|---|
| Shikra-7B [12] | 49.7 | 19.8 | 43.6 | 11.4 | – | – |
| GPT4RoI-7B [84] | 51.3 | 12.0 | 48.0 | 12.1 | – | – |
| Osprey-7B [80] | 65.2 | 38.2 | 73.1 | 52.7 | – | – |
| Ferret-13B [77] | 65.0 | 37.8 | – | – | – | – |
| VP-LLAVA-8B [41] | 86.7 | 61.5 | 75.7 | 50.0 | 44.8 | 46.9 |
| VP-SPHINX-13B [41] | 87.1 | 62.9 | 76.8 | 51.3 | **45.4** | 48.8 |
| DAM-8B [38] | **89.0** | 77.7 | 84.2 | 73.2 | – | – |
| **PAM-1.5B (Ours)** | 87.4 | 76.5 | 85.1 | 73.5 | 39.4 | 48.6 |
| **PAM-3B (Ours)** | 88.6 | **78.3** | **87.4** | **74.9** | 42.2 | **52.3** |

Table 1: Results of region-level image recognition on LVIS, PACO, COCO Text, and Total-Text.

| Model | VG | | Refcocog | | Ref-L4 | | | Ferret Bench | MDVP Bench |
| | METEOR | CIDEr | METEOR | CIDEr | ROUGE-L | METEOR | CIDEr | Refer. Desc. | Avg. |
|---|---|---|---|---|---|---|---|---|---|
| GLaMM-7B [51] | 17.0 | 127.0 | 15.7 | 104.0 | 23.8 | 10.1 | 51.1 | - | - |
| Osprey-7B [80] | - | - | 16.6 | 108.3 | - | - | - | 72.2 | 44.3 |
| Ferret-7B [77] | - | - | - | - | 22.3 | 10.7 | 39.7 | 68.7 | 47.6 |
| VP-LLaVA-8B [41] | - | - | 22.4 | 153.6 | - | - | - | 75.2 | 70.6 |
| VP-SPHINX-13B [41] | 20.6 | 141.8 | 23.9 | **162.5** | 22.6 | 10.7 | 32.4 | 77.4 | **74.3** |
| Omni-RGPT-7B [25] | 17.0 | 139.3 | 17.0 | 109.7 | - | - | - | - | - |
| RegionGPT-7B [21] | 17.0 | 145.6 | 16.9 | 109.9 | 25.3 | 12.2 | 42.0 | - | - |
| DAM-8B [38] | - | - | - | - | 37.1 | 19.4 | 70.0 | - | - |
| **PAM-1.5B (Ours)** | 19.2 | 132.9 | 24.7 | 135.0 | 29.6 | 15.9 | 55.8 | 75.4 | 69.4 |
| **PAM-3B (Ours)** | 20.8 | 142.3 | 26.9 | 143.1 | 31.3 | 17.2 | 59.7 | **78.3** | 72.2 |

Table 2: Performance comparison on region-level image captioning across multiple benchmarks.

SoM. *(4) LLM-Powered Elaboration*: Then, using the original annotations as condition, we prompt GPT-4o to generate descriptions that are both refined, detailed and temporally aware. This multi-frame consolidation is critical as it enhances GPT-4o's contextual comprehension, yielding superior descriptions compared to individual frame analysis.

**Region-level Streaming Video Caption.** Beyond describing the entire video, we aim to extend the model's capabilities to a streaming manner. To achieve this, we perform additional augmentation on our refined region-level video caption data. Specifically, we first employ the TRACE-Uni model [22] to segment the input video into multiple distinct events, each demarcated by its temporal boundaries. Subsequently, for each segmented video clip, we apply the same 'storyboard-driven' processing method. To generate precise and continuous event descriptions, the GPT-4o input prompt was redesigned to iteratively incorporate the description from the preceding video clip as contextual information for processing the current clip. The entire workflow is illustrated in Fig. 5(right).

# 5 Experiments

## 5.1 Implementation Details

We employ Qwen2.5-1.5B/3B [72] as our semantic decoder, and utilize the pre-trained hierarchical SAM 2-Large[3] as the base vision foundation model. By default, we use 16 learnable semantic tokens and uniformly sample 16 frames per video clip. All training is conducted on 8 NVIDIA A100 GPUs with 80GB. For all evaluation experiments, we adopt a zero-shot test manner without fine-tuning on specific datasets. The best and the second best results are indicated in **bold** and with underline

## 5.2 Image Benchmarks

**Regional Recognition and Explanation.** This task requires the model to identify either the object category or scene text within a specified image region. Recognition performance is assessed on the validation sets of the LVIS (object-level) [23] and PACO (part-level) [50] datasets, alongside the test sets of COCO-Text [61] and Total-Text [14]. Standard evaluation metrics include Semantic Similarity [80], Semantic Intersection over Union (Sem. IoU) [15], and accuracy.

---

[3]https://dl.fbaipublicfiles.com/segment_anything_2/092824/sam2.1_hiera_large.pt

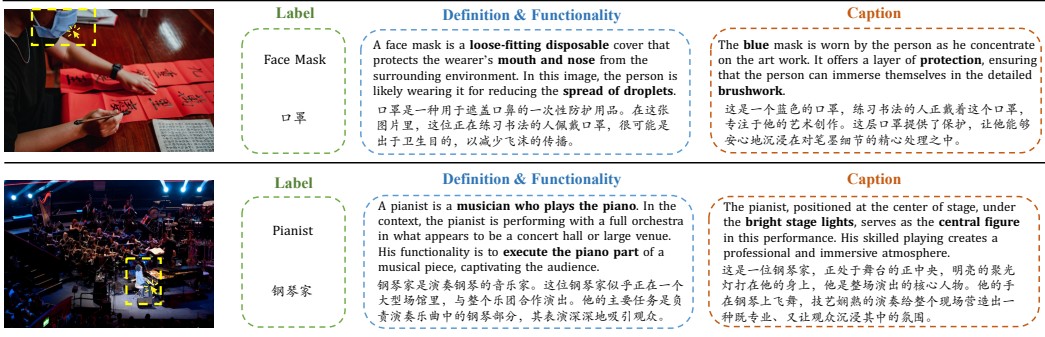

Figure 6: PAM provides various semantic granularities informantion and support bilingual outputs.

As shown in Table 1, both our PAM-1.5B and PAM-3B demonstrate strong performance. Notably, PAM-3B significantly outperforms other competing methods. It achieves optimal performance on the PACO benchmark, exceeding the previous best model by over 3.2%, and surpasses the current SOTA model, DAM-8B, on the LVIS benchmark in terms of semantic IoU. Furthermore, as indicated in the right column of Table 1, our PAM-3B outperforms VP-SPHINX-13B by over 3.5% on Total-Text and achieves comparable performance on COCO-Text. These results demonstrate its promising capabilities in scene text recognition. We further showcase qualitative visualizations in Fig. 6, illustrating PAM's effectiveness in generating insightful explanations that cover both the general definition and the contextual role of prompted objects.

**Regional Caption.** We evaluate the model's capability to generate both concise and detailed region descriptions on multiple benchmarks. For concise region captioning, we evaluate on the validation splits of RefCOCOg [32] and Visual Genome (VG)[36]. For more expressive descriptions, assessments are conducted on the challenging Ref-L4[10] dataset. Caption quality is measured using ROUGE-L [39], METEOR [3], and CIDEr [60]. Additionally, we benchmark referring descriptions via Ferret-Bench [77] and MDVP-Bench [41], where GPT-4o is employed to gauge the quality of the generated responses.

As the results shown in Table 2, PAM-3B surpasses existing methods on the VG, RefCOCOg, and Ferret benchmarks. On MDVP-Bench, it achieves performance comparable to the current SOTA method, VP-SPHINX-13B. Furthermore, on the Ref-L4 benchmark, PAM-3B demonstrates outstanding performance, surpassing all models except the top-performing DAM-8B. Notably, these competitive results are achieved with fewer parameters and reduced computational cost, highlighting PAM's excellent balance of performance and efficiency.

## 5.3 Video Benchmarks

| Model | Elysium | BensMOT | HC-STVG | | VideoRefer-Bench-D | | | | |
|---|---|---|---|---|---|---|---|---|---|
| | METEOR | METEOR | METEOR | CIDEr | SC | AD | TD | HD | Avg. |
| Elysium-7B [63] | 19.1 | 1.1 | – | – | 2.35 | 0.30 | 0.02 | 3.59 | 1.57 |
| Merlin-7B [78] | – | – | 11.3 | 10.5 | – | – | – | – | – |
| Omni-RGPT-7B [25] | 9.3 | 14.6 | – | – | – | – | – | – | – |
| Artemis-7B [49] | – | – | 18.0 | 53.2 | 3.42 | 1.34 | 1.39 | 2.90 | 2.26 |
| VideoRefer-7B [81] | – | – | 18.7 | 68.6 | 4.44 | 3.27 | 3.10 | 3.04 | 3.46 |
| DAM-8B [38] | - | - | 21.0 | **91.0** | **4.69** | **3.61** | **3.34** | **3.09** | **3.68** |
| **PAM-1.5B (ours)** | 22.7 | 20.1 | 19.8 | 65.9 | 3.73 | 2.75 | 2.77 | 2.89 | 3.03 |
| **PAM-3B (ours)** | **24.3** | **21.6** | **23.3** | _70.3_ | 3.92 | 2.84 | 2.88 | 2.94 | 3.14 |

Table 3: Performance comparison on video region captioning.

| Model | ActivityNet | | |
|---|---|---|---|
| | CIDEr | METEOR | G-STDC |
| VideoRefer-7B [81] | 22.1 | 14.7 | 1.73 |
| DAM-3B [38] | 11.3 | 14.8 | 0.94 |
| GIT* [65] | 29.8 | 7.8 | – |
| Vid2Seq* [73] | 30.2 | 8.5 | – |
| Streaming Vid2Seq* [73] | **37.8** | 10.0 | – |
| **PAM-1.5B (ours)** | 28.6 | 24.8 | 2.43 |
| **PAM-3B (ours)** | _30.1_ | **27.3** | **2.67** |

Table 4: Performance for streaming region captioning on ActivityNet.

**Video Region Caption.** This task requires the model to generate an accurate and temporally-aware description for a prompted region within the video's context. We primarily evaluate on four public benchmarks: Elysium [63], BensMOT [37], HC-STVG [59], and VideoRefer-Bench-D [81]. As shown in Table 3, our PAM-1.5B and PAM-3B achieve the SOTA performance on both the Elysium and BensMOT benchmarks. Furthermore, our PAM-3B surpasses the current SOTA method, DAM-8B, by 2.3% in terms of METEOR on the HC-STVG benchmark. On the VideoRefer-Bench, our

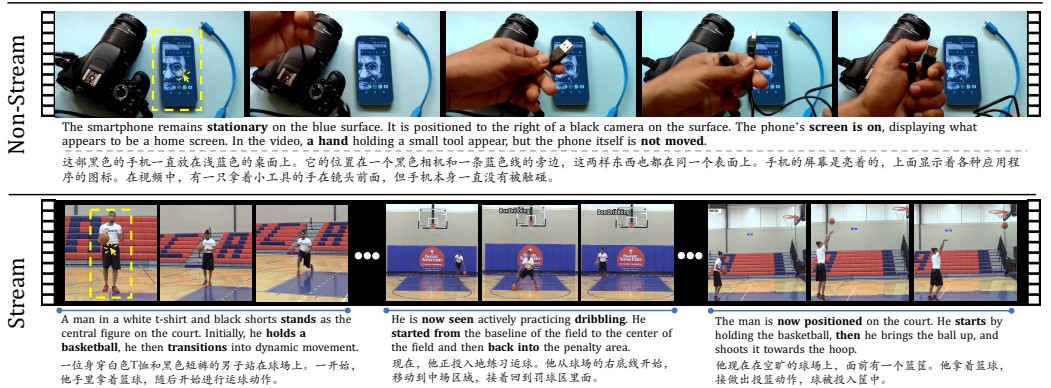

Figure 7: Qualitative visualization examples of PAM for region-level non-streaming and streaming video caption.

models exhibit marginally lower performance compared to VideoRefer-7B and DAM-8B, indicating potential for further improvement.

**Streaming Video Region Caption.**    This task requires the model to generate continuous descriptions for a prompted region in a streaming manner. For evaluation, we primarily utilize the validation set of the ActivityNet dataset [7]. To ensure a fair comparison and to accurately assess region-level streaming captioning capabilities, we manually curated a subset of 400 samples. This selection process adhered to two key criteria:

*(1)* each annotated event within a given video is temporally continuous and non-overlapping, and *(2)* all annotated event descriptions for that video pertain to the same subject. Subsequently, we manually annotated bounding boxes for the target subject in each selected video. We initially employ two standard dense captioning metrics for evaluation: CIDEr and ME-TEOR. To further assess the continuity and entity consistency of descriptions for sequential events, we propose a new metric: the

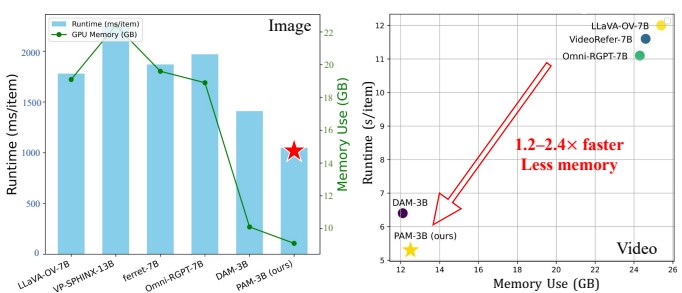

Figure 8: Comparison of GPU memory usage and inference efficiency on an A6000 GPU.

GPT-4o-evaluated Spatio-Temporal Description Continuity Score (G-STDC), which ranges from 0 to 5. (Details in App. C). The results in Table 4 indicate that recent region-level video caption models, including VideoRefer and DAM, exhibit limited capability in the streaming caption task. Compared to general streaming caption approaches such as Streaming Vid2Seq, our PAM-3B outperforms it on the METEOR metric. Furthermore, PAM-3B achieves optimal performance on G-STDC, indicating its excellent spatio-temporal continuity and ability to maintain consistent subject descriptions.

## 5.4 Efficiency

As shown in Fig. 8, compared to existing works, our PAM demonstrates superior inference efficiency and requires less GPU memory for both image and video processing, highlighting its suitability for efficient deployment in real-world applications.

## 5.5 Ablations

We study the effectiveness of the proposed key techniques as below.

| Method | LVIS (S.IoU) | RefCOCOg (METEOR) | HC-STVG (METEOR) | time (ms/it) |
|---|---|---|---|---|
| + 4 sem.T | 78.9 | 26.1 | 22.5 | 972 |
| + 16 sem.T | 79.6 | 26.9 | 23.3 | 980 |
| + 64 sem.T | **80.0** | **27.0** | **23.5** | 1143 |
| + w/o sem.T | 77.6 | 24.6 | 21.3 | **967** |

Table 5: Number of Sem.T.

| Method | LVIS (S.IoU) | RefCOCOg (METEOR) | HC-STVG (METEOR) |
|---|---|---|---|
| All in one | 78.7 | 25.8 | 21.6 |
| S1→2 | **79.7** | 26.7 | 22.4 |
| S1→1.5 →2 | 79.6 | **26.9** | **23.3** |

Table 6: Different training stage.

| Method | LVIS (S.IoU) | RefCOCOg (METEOR) | HC-STVG (METEOR) |
|---|---|---|---|
| I.E. pre S2-FFM | 78.4 | 25.0 | 21.9 |
| I.E. after S2-FFM | **79.6** | **26.9** | **23.3** |
| all T. + sem.T | 79.9 | 26.8 | 23.3 |
| mask T. + sem.T | 79.6 | **26.9** | 23.3 |

Table 7: Impact of different intermediate embeddings.

• In Table 5, we present the impact of adjusting the number of learnable semantic tokens (sem.T). It is observed that using an insufficient number of sem.T leads to a drop in performance. Conversely, using an excessive number of sem.T results in diminishing gains, while also increasing the computational cost. Therefore, we select 16 sem.T to achieve a favorable performance-efficiency trade-off.

• In Table 6, we compare different training strategies. It is seen that initialization from the image-video model checkpoint (from Stage 1.5) consistently leads to enhanced performance compared to either initializing directly from a Stage 1 model checkpoint or training directly in an all-in-one stage.

• Table 7 compares the impact of different intermediate features from SAM 2. The results show that embeddings updated by S2-FFM enhance our model's performance, which further underscore the critical role of the feature selection approach.

## 6   Conclusion

We present Perceive Anything Model (PAM), a region-level vision-language model extended from SAM 2, designed for simultaneous segmentation of objects while generating diverse semantic outputs of them across both images and videos. PAM demonstrates robust performance on multiple region-level understanding tasks while achieving high computational efficiency. The simplicity and efficiency of our approach make it well-suitable for real-world applications, enabling a fine-grained, multi-dimensional understanding of visual content from a single interaction.

## Acknowledgments

This study was supported in part by National Key R&D Program of China Project 2022ZD0161100, in part by the Centre for Perceptual and Interactive Intelligence, a CUHK-led InnoCentre under the InnoHK initiative of the Innovation and Technology Commission of the Hong Kong Special Administrative Region Government, in part by NSFC-RGC Project N_CUHK498/24, and in part by Guangdong Basic and Applied Basic Research Foundation (No.2023B1515130008, XW).

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

# Appendix

## A  Configuration for Each Training Stage

Table 8 details the configurations for each training stage of the Perceive Anything Model (PAM). It outlines the vision parameters, dataset characteristics, model specifications, and training hyperparameters throughout the curriculum learning stages. The maximum number of visual tokens varies by input modality: single images are represented using 1024 tokens, while for videos, we sample up to 16 frames, leading to a maximum of 4864 visual tokens. A global batch size of 1024 is used for stages 1 and 1.5, and 256 for stage 2.

|  | Stage 1 | Stage 1.5 | Stage 2 |
|---|---|---|---|
| **Visual Tokens** | 1024 | 1024 + N×256
MAX 1024 + 15×256 = 4864 | 1024 + N×256
MAX 1024 + 15×256 = 4864 |
| **Sem. Tokens** | 16 | N×16 (MAX 256) | N×16 (MAX 256) |
| **Dataset** | image classification
image caption | image classification
image caption
video caption | image classification
image explanation
video caption
streaming video caption |
| **Trainable components**
# 1.5B
# 3B | Sem. Perceiver + Projector
7.6M
7.7M | Sem. Perceiver + Projector
7.6M
7.7M | Sem. Perceiver, Projector, LLM
1.6B
3.1B |
| **Batch Size**
**Learning Rate**
**Epoch**
**Warmup Ratio**
**Optimizer** | 1024
$1\times10^{-4}$
1
0.03
AdamW | 1024
$4\times10^{-5}$
1
0.03
AdamW | 256
$1\times10^{-5}$
1
0.03
AdamW |

Table 8: Detailed configuration for each training stage of the PAM.

## B  Dataset

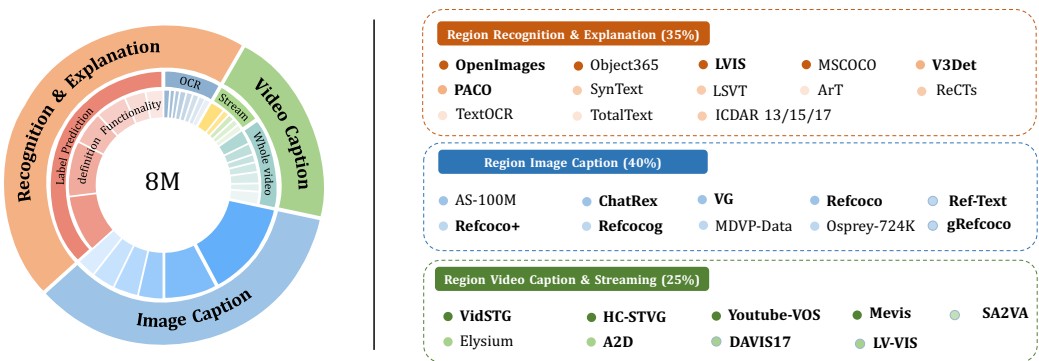

Figure 9: All Public Dataset Collection. Datasets highlighted in **bold** are selected for further refinement and augmentation pipeline, aimed at generating a high-quality training dataset.

### B.1  More Details of Image Data Construction

This section details our image data construction process. To generate data encompassing clear conceptual explanations, contextual functional roles, and detailed descriptions of specific regional capabilities, we primarily leveraged the extensive world knowledge of leading VLMs.

Our approach involved several stages. Initially, we collected data from public referring object detection, segmentation, and captioning datasets [32, 6, 43, 36, 29, 68]. While these sources provide

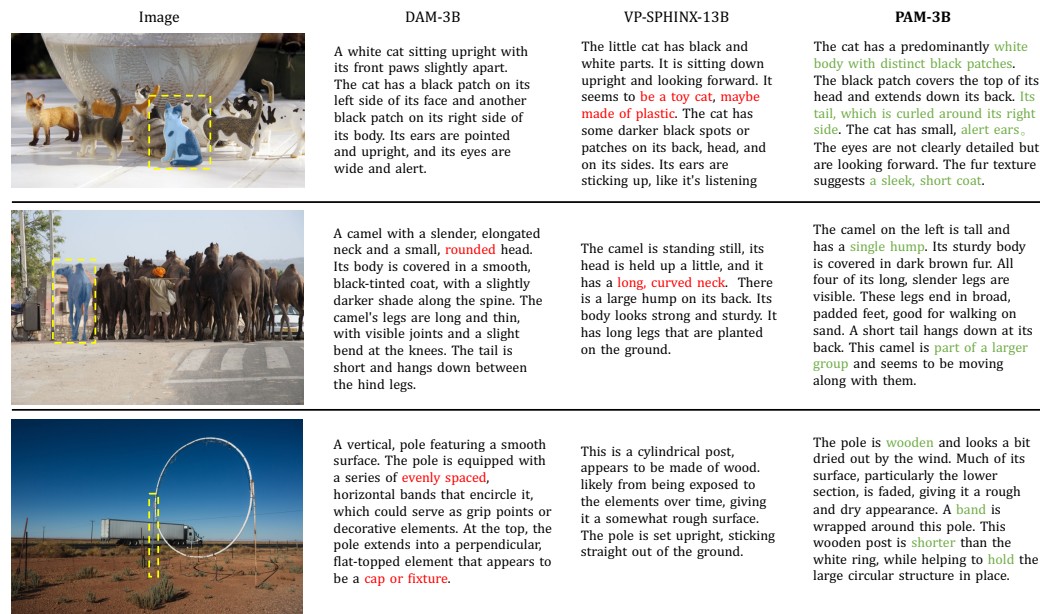

| Image | DAM-3B | VP-SPHINX-13B | **PAM-3B** |
|---|---|---|---|
| | A white cat sitting upright with its front paws slightly apart. The cat has a black patch on its left side of its face and another black patch on its right side of its body. Its ears are pointed and upright, and its eyes are wide and alert. | The little cat has black and white parts. It is sitting down upright and looking forward. It seems to be a toy cat, maybe made of plastic. The cat has some darker black spots or patches on its back, head, and on its sides. Its ears are sticking up, like it's listening | The cat has a predominantly white body with distinct black patches. The black patch covers the top of its head and extends down its back. Its tail, which is curled around its right side. The cat has small, alert ears. The eyes are not clearly detailed but are looking forward. The fur texture suggests a sleek, short coat. |
| | A camel with a slender, elongated neck and a small, rounded head. Its body is covered in a smooth, black-tinted coat, with a slightly darker shade along the spine. The camel's legs are long and thin, with visible joints and a slight bend at the knees. The tail is short and hangs down between the hind legs. | The camel is standing still, its head is held up a little, and it has a long, curved neck. There is a large hump on its back. Its body looks strong and sturdy. It has long legs that are planted on the ground. | The camel on the left is tall and has a single hump. Its sturdy body is covered in dark brown fur. All four of its long, slender legs are visible. These legs end in broad, padded feet, good for walking on sand. A short tail hangs down at its back. This camel is part of a larger group and seems to be moving along with them. |
| | A vertical, pole featuring a smooth surface. The pole is equipped with a series of evenly spaced, horizontal bands that encircle it, which could serve as grip points or decorative elements. At the top, the pole extends into a perpendicular, flat-topped element that appears to be a cap or fixture. | This is a cylindrical post, appears to be made of wood. likely from being exposed to the elements over time, giving it a somewhat rough surface. The pole is set upright, sticking straight out of the ground. | The pole is wooden and looks a bit dried out by the wind. Much of its surface, particularly the lower section, is faded, giving it a rough and dry appearance. A band is wrapped around this pole. This wooden post is shorter than the white ring, while helping to hold the large circular structure in place. |

Figure 10: Qualitative comparison between PAM and prior models.

unique descriptions for each region, they often lack comprehensive semantic details. Therefore, our work focused on refining and augmenting these data to achieve richer semantic granularity. Specifically, the Set-of-Mark (SoM) prompting method [75] was initially employed to identify regions of interest within the images. To generate high-quality conceptual explanations and contextual functionalities for these identified regions, we manually refined the prompts and then utilized strong closed-sourced model, GPT-4o [27], to produce the desired textual responses. For crafting detailed descriptions of these regions, a powerful open-sourced model, Qwen2.5-VL-72B [2], was employed to expand and supplement existing textual information. Following this automated expansion, a two-stage cleaning process was implemented. First, rule-based methods were applied for preliminary filtering, addressing issues such as output format inconsistencies. Subsequently, manual review was conducted to identify and isolate remaining inaccurate or low-quality data, which were then re-annotated.

## B.2 Bilingual Annotations

To support bilingual (English and Chinese) output, we extended our refined datasets by generating corresponding Chinese versions. For the majority of these datasets, Chinese annotations were directly created using Chinese prompts. In cases where data was initially generated with GPT-4o, the existing English content was translated into Chinese utilizing the DeepSeek-V3 model [42].

## C  Implement Details of G-STDC

In Sec. 5.3, we introduce the GPT-4o-evaluated Spatio-Temporal Description Continuity Score (G-STDC) to assess the continuity and entity consistency of descriptions for sequential events. Specifically, for a given video, the predicted descriptions and the corresponding ground truth descriptions for its multiple events are first sorted chronologically. Both sequences are then provided to GPT-4o for evaluation. During this process, GPT-4o assesses the predicted descriptions based on temporal continuity and entity consistency (in relation to the ground truths), assigning a G-STDC score on a scale of 0 to 5, where 5 represents the optimal score and 0 the poorest. Results are presented in Table 4.

# D    More Analysis of PAM

## D.1    Performance for Background

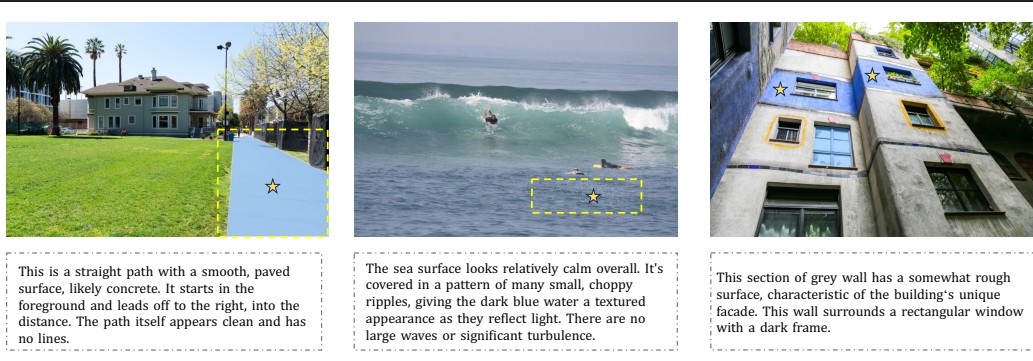

Figure 11: PAM can accurately describe specific background areas, such as roads, ground surfaces, sea, walls, the sky, and more.

## D.2    Failure Cases

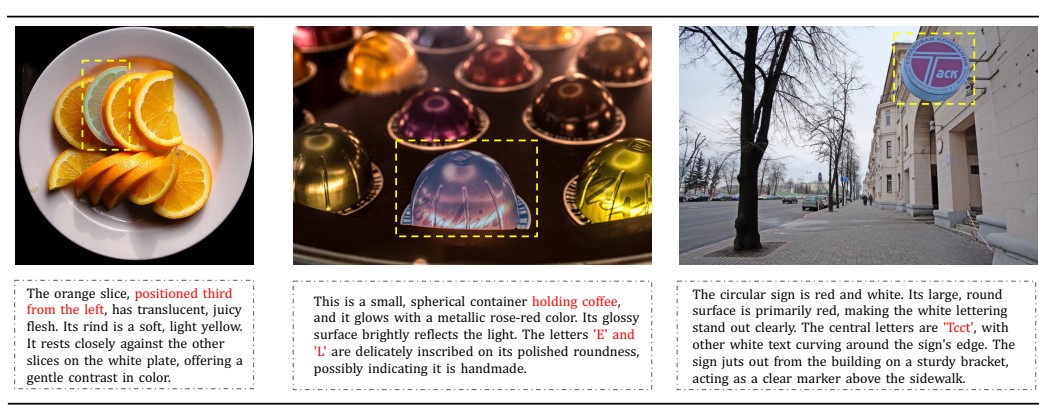

Figure 12: Failure Cases of PAM in Images.

Fig. 12 illustrates several of PAM's failure cases. As these examples show, PAM sometimes makes errors in its descriptions: (1) For instance, in the first image, the orange slice is the second one, but PAM describes it as the third. (2) In the third image, the actual text is 'Tack', but PAM reads it as 'Tcct'. (3) PAM occasionally describes elements not present in the image, such as describing engraved letters in the second image, which has no such features.

As shown in Fig. 13, PAM also exhibits certain limitations in video contexts. Specifically, if a user-specified object occupies a minor portion of the frame and temporarily disappears from view, PAM may default to describing the most salient object in the scene instead. We attribute this behavior to potential biases in the training data: the GPT-assisted method employed for annotation during data construction might favor describing the most salient object over the specific region, thereby introducing label inaccuracies. Additionally, in streaming captioning, PAM might be influenced by historical descriptions, leading its output for the current video clip to be overly similar to that of the preceding clip.

We expect these errors to be mitigated by broader data coverage and further reinforcement training.

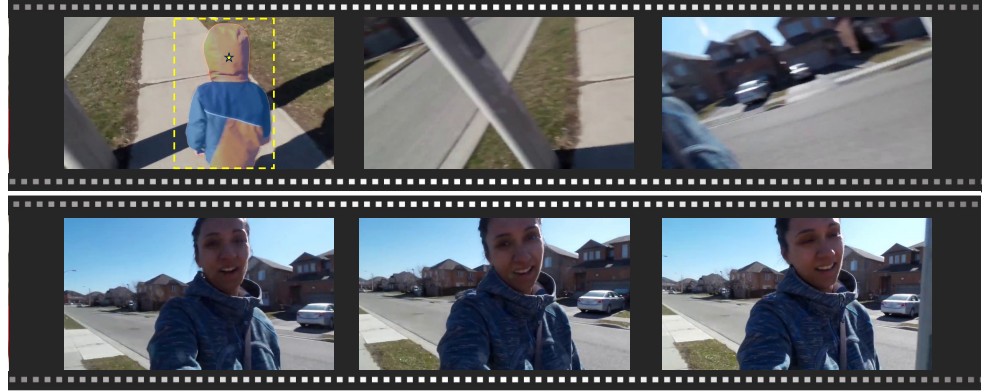

A woman walking down a relatively empty street while holding her phone. She is wearing a blue jacket and appears to be taking a selfie or recording a video with her phone, as her head turns from time to time. Her expression is quite calm.

Figure 13: Failure Cases of PAM in Videos.

### D.3 Performance on Long Videos

PAM's performance on long videos depends on the number of video frames processed. A larger number of frames enables PAM to generate more information-rich descriptions, but this incurs an exponential increase in computational cost. With its default setting of sampling 16 frames, PAM can only provide broad and coarse descriptions of the states and changes of specific subjects in long videos. However, PAM features a region-level streaming captioning capability that can improve its handling of long videos. This method involves segmenting a long video into multiple shorter clips; descriptions are then generated for each clip sequentially and subsequently merged to create a single, detailed description of the entire video.

## E   Potential Limitations and Discussions

**Limited Capability for General Understanding Tasks.**    PAM is currently trained for a specific set of four region-level understanding tasks: category prediction, brief and detailed regional captioning, video captioning, and streaming region captioning. Therefore, it presently lacks support for other general vision-language tasks, such as Visual Question Answering (VQA). However, the architecture and training strategy of PAM are inherently well-suited to accommodate these broader functionalities. Looking ahead, we plan to develop additional high-quality conversational datasets to extend PAM's capabilities to encompass both region-level image and video dialogue.

**Limitations in Real-Time Streaming Video Region Captioning.**    Despite being 1.2–2.4$\times$ faster than existing models, PAM's capability for real-time streaming video region captioning is currently hindered by the excessive number of visual tokens requiring processing by the LLM. In the future, we aim to further identify methods for substantially reducing the number of visual tokens, with the goal of achieving real-time efficiency while maintain the robust understanding performance.

