# OpenReview forum: "Perceive Anything: Recognize, Explain, Caption, and Segment Anything in Images and Videos"
_NeurIPS.cc/2025/Conference — NeurIPS 2025 poster_

### Official Review · Reviewer_R2WL · 2025-06-22

**Clarity:** 3
**Significance:** 3
**Originality:** 3
**Rating:** 4
**Confidence:** 4

**Summary:**

This paper introduces the Perceive Anything Model (PAM), a vision-language model designed for comprehensive region-level understanding in images and videos. PAM builds on the Segment Anything Model 2 (SAM 2) and integrates Large Language Models (LLMs) to achieve simultaneous object segmentation and diverse semantic output generation.  Compared to previous methods, PAM improvements in inference efficiency and GPU memory usage.

**Questions:**

- Data Processing using External Models

The authors mention using advanced VLMs such as qwen2.5-vl-72B, gpt4o, and deepseek v3 for data labeling and processing. The performance gains of PAM may not be attributed to its architectural design, but could be significantly determined by the power of these external models and the extensive labeled data utilized. Thus, the true effectiveness of PAM's own structure remains a question.

- Comparison with Existing VLMs

The authors seem to avoid direct comparison of PAM with those existing VLMs such as open-sourced Qwen2.5-VL series. Although the author mentioned the differences between region-level VLMs and image-level VLMs, I think that these VLMs can be adapted to perform region-level tasks with the help of system prompts and few-shot examples. A fair comparison with these VLMs may be important. Since if existing VLMs are already capable of handling these tasks, the necessity of distilling a new region-level VLM appears to be questionable.

- Explanation of Efficiency Gains

The authors highlight PAM's superior inference efficiency and reduced GPU memory usage for image and video processing. However, they do not provide a sufficiently detailed explanation of the sources of these efficiency improvements. It is unclear whether the faster speed is primarily due to the smaller model size (e.g., 3B vs. 7B), different model architecture, or other factors such as more efficient processing pipelines. A clear analysis may be better.

**Ethical Concerns:**

["NO or VERY MINOR ethics concerns only"]

**Final Justification:**

The author's response solves my concern. I keep my rating as 4: Borderline accept.

**Limitations:**

yes

**Quality:**

3

**Strengths And Weaknesses:**

Strengths:

- The integration of SAM 2 with LLMs via the Semantic Perceiver is a successful attempt to combining segmentation with rich semantic understanding, enabling multi-dimensional outputs like region-level captions and streaming video captioning. The performance and ablations show the power of the proposed method.
- The video data refinement pipeline, storyboard-driven caption expansion, addresses a critical need and efficient way for fine-grained semantic annotations.

Weakness:

See questions.

---

> ### Author Rebuttal · Authors · 2025-07-31
>
> ## **Response to W1: Performance gains of PAM may not be attributed to its architectural design，but rather from its utilization of extensive labeled data**
>
> Data is one of the key contributions of our work. We identified shortcomings in open-source data annotations and performed a comprehensive refinement. We acknowledge that this refined data provided a significant boost to PAM’s performance.
>
> However, for the classification and OCR tasks in Table 1, we used the original open-source data. On these benchmarks, our PAM-3B model achieves SOTA results on LVIS (semantic IoU), PACO, and Total-Text. Notably, all competing models we compared against are 7B or larger in scale. This demonstrates the effectiveness of the PAM architecture itself, independent of the data enhancements.
>
> ## **Response to W2: Comparison with Existing general VLMs.**
>
> Thank you for pointing this out. To demonstrate the necessity of region-level VLMs, we compared them against general VLMs on the DLC-Bench[1], which evaluates accuracy on detailed localized captioning. The experimental results are as follows:
>
> | Method              | Pos (%) | Neg (%) | Avg(%) |
> |:--------------------|:-------:|:-------:|:------:|
> | **General VLMs**    |         |         |        |
> | Llama-3.2-V-11B     | 30.7    | 63.8    | 47.3   |
> | InternVL2.5-8B      | 15.9    | 42.0    | 28.9   |
> | Qwen2.5-VL-7B       | 20.3    | 62.2    | 41.2   |
> | **Region-level VLMs** |         |         |        |
> | VP-SPHINX-13B       | 26.3    | **71.6**    | 49.0   |
> | PAM-3B (ours)          | **34.5**    | 63.1    | 48.8   |
>
> As the results indicate, Region-level VLMs significantly outperform general VLMs on fine-grained detailed captioning tasks. This validates the necessity of research into region-level VLMs.
>
> [1] Describe Anything: Detailed Localized Image and Video Captioning
>
>
> ## **Response to W3: Explanation of Efficiency Gains**
>
> | Model/Img. | Runtime (ms/item) | Memory Use (GB) |
> |:-----------------|:-----------------:|:---------------:|
> | Ferret-7B          | 1845             | 18.7            |
> | Omni-RGPT-7B     | 1970             | 20.8            |
> | PAM-7B           | **1615**          |  **17.1**      |
>
> Thank you for your insightful comments. To better demonstrate the efficiency of our model, we trained a PAM-7B to compare it directly with other 7B models. This approach allows us to rule out the influence of model size and focus on architectural advantages.
>
> The experimental results show that PAM-7B still outperforms others in terms of efficiency. This superiority can be attributed to three key factors:
>
> 1.	SAM 2, which serves as PAM’s vision encoder, is more lightweight compared to the encoders used in other VLMs.
>
> 2.	Fewer Visual Tokens: The number of visual tokens processed from SAM 2 and fed into the LLM is significantly lower (1024 vs. more than 2048) than in other models.
>
> 3.	PAM utilizes the latent features from SAM 2, so it does not need to wait for the entire SAM 2 inference process to complete. Once the latent features are obtained, mask decoding and semantic decoding can proceed in parallel.

---

> ### Author Response · Authors · 2025-08-04
>
> Dear Reviewer R2WL,
>
> Many thanks for your time in reviewing our paper and your constructive comments. We have submitted the point-to-point responses on **attribution of performance gains**, **comparison with general VLMs**, and **explanation of efficiency gains**. We would greatly appreciate it if you could let us know whether your concerns have been addressed. We are also happy to provide further clarifications if needed.
>
> Best regards,
>
> Authors of paper #3605 *Perceive Anything: Recognize, Explain, Caption, and Segment Anything in Images and Videos*

---

> > ### Comment · Reviewer_R2WL · 2025-08-05
> >
> > Thanks for the authors' detailed response. My concerns has been fully solved, thus I keep my rating as 4: Borderline accept.

---

> > > ### Author Response · Authors · 2025-08-06
> > > **Thanks for your recognition of our rebuttal!**
> > >
> > > Dear Reviewer R2WL,
> > >
> > > Thank you for acknowledging our rebuttal and efforts! We deeply appreciate your insightful comments, which have been invaluable in helping us improve our work.
> > >
> > > Regards,
> > >
> > > PAM Authors

---

### Official Review · Reviewer_y2LG · 2025-07-02

**Clarity:** 3
**Significance:** 3
**Originality:** 3
**Rating:** 4
**Confidence:** 4

**Summary:**

This paper focuses on the vision language model area and introduces an end-to-end vision-language model, PAM, to bridge the gap between object segmentation and deep semantic understanding in both images and videos. This work also introduces a large-scale, high-quality dataset comprising 1.8M image and 0.6M video samples, which features multi-level semantic granularity. The authors conducted comprehensive experiments and validate the effectiveness of the proposed architecture and dataset.

**Questions:**

See weaknesses

**Ethical Concerns:**

["NO or VERY MINOR ethics concerns only"]

**Quality:**

3

**Strengths And Weaknesses:**

Strengths:
1. The authors successfully integrate segmentation with deep and multi-level semantic understanding, which is a significant step towards more holistic machine perception.
2. The authors have made a substantial contribution by curating a large and high-quality dataset. The dataset's multi-dimensional annotations (category, definition, function, caption) and its bilingual support are valuable.
3. The model's capabilities are thoroughly validated across a wide array of demanding benchmarks for both images (LVIS, PACO, RefCOCOg, Ferret-Bench) and videos (Elysium, BensMOT, HC-STVG). The authors also propose a new, relevant metric, G-STDC, for evaluating the continuity of streaming captions, demonstrating a thoughtful approach to assessment.

Weaknesses
1. The authors transparently acknowledge that PAM is specialized for a specific set of region-level understanding tasks. It currently lacks support for more general vision-language tasks like open-ended Visual Question Answering (VQA) or region-level dialogue, which limits its scope compared to some other large multimodal models.
2. The experiments and ablation studies in this work were primarily conducted on Qwen2.5-1.5B/3B. This relatively limited selection raises concerns about the generalizability and scalability of the method.

---

> ### Author Rebuttal · Authors · 2025-07-31
>
> ## **Response to W1: Lacks support for more general vision-language tasks like open-ended (VQA)**
>
> Although PAM specializes in providing accurate and detailed region-level descriptions, its core architecture inherently supports other understanding tasks, such as visual question answering (VQA), without any modification.
>
> To demonstrate this capability, we trained our model on several open-source, region-level VQA datasets, including MDVP-Instruct, VCR, Visual7W, and Osprey-724K. The training data was formatted as question-answer pairs targeting a specific area, for example: “What kind of clothes is the man wearing in this \<region\>?”
>
> The experimental results are as follows:
>
> | | **Ferret Bench** |  | **VCR** | |
> | :--- | :---: | :---: | :---: | :---: |
> | | Reasoning | Q -> A (%) | QA -> R (%) | Q -> AR (%) |
> | Osprey-7B | 67.8 | - | - | - |
> | Ferret-7B | 67.3 | - | - | - |
> | VILLA-L | - | 78.4 | 82.5 | 65.1 |
> | GPT4RoI-7B | - | 87.4 | **89.6** | 78.6 |
> | **PAM-3B (ours)**| **68.1** | **87.9** | 89.3 | **79.1** |
>
> ## **Response to W2: Concerns about the generalizability and scalability**
>
> Thank you for pointing this out. Our initial objective was to train a region-level VLM that is fast, compact, and high-performing, making it easier to deploy in practical, real-world scenarios. At the same time, we recognize that scalability is crucial. Therefore, we dedicated additional time and resources to train PAM-7B. The experimental results are as follows. As can be observed, PAM-7B exhibits outstanding performance, which demonstrates that our method possesses robust generalizability and scalability. Looking ahead, if sufficient computational resources are available, we will involve testing a 32B model to further evaluate its scalability.
>
> | | LVIS (Img Cls.) | | Ref-L4 (Img Cap.) | | | VideoRefer (Video Cap.) |
> | :--- | :---: | :---: | :---: | :---: | :---: | :---: |
> | | Sem. Sim. | Sem. IoU | ROUGE-L | METEOR | CIDEr | Avg. |
> | Ferret-7B | 65.0 | 37.8 | 22.3 | 10.7 | 39.7 | - |
> | VP-SPHINX-13B | 87.1 | 62.9 | 22.6 | 10.7 | 32.4 | - |
> | VideoRefer-7B | - | - | - | - | - | 3.46 |
> | **PAM-3B** | 88.6 | 78.3 | 31.1 | 17.2 | 59.7 | 3.14 |
> | **PAM-7B** | **88.9** | **80.1** | **36.5** | **19.1** | **65.9** | **3.50** |

---

> ### Author Response · Authors · 2025-08-04
>
> Dear Reviewer y2LG,
>
> Many thanks for your time in reviewing our paper and your constructive comments. We have submitted the point-to-point responses on **the ability to more general vision-language tasks**, and **the generalizability and scalability**. We would greatly appreciate it if you could let us know whether your concerns have been addressed. We are also happy to provide further clarifications if needed.
>
> Best regards,
>
> Authors of paper #3605 *Perceive Anything: Recognize, Explain, Caption, and Segment Anything in Images and Videos*

---

> ### Comment · Reviewer_y2LG · 2025-08-05
> **Response to Authors**
>
> The authors have solved most of my questions. Therefore, I choose to keep my score.

---

> > ### Author Response · Authors · 2025-08-06
> > **Thanks for your recognition of our rebuttal!**
> >
> > Dear Reviewer y2LG,
> >
> > Thank you for acknowledging our rebuttal and efforts! We deeply appreciate your insightful comments, which have been invaluable in helping us improve our work.
> >
> > Regards,
> > PAM Authors

---

### Official Review · Reviewer_H65M · 2025-07-02

**Clarity:** 3
**Significance:** 2
**Originality:** 3
**Rating:** 4
**Confidence:** 4

**Summary:**

This paper proposes Percieve Anything Model (PAM), which integreates the visual understanding ability of SAM2 into LLMs. Specifically, a frozen SAM2 is connected with LLM with trainable semantic perciever. While the segmentation ability of PAM is fully inherented from SAM2, such integretion provides detailed visual information for LLMs to perform recognition, explaination and caption. To train such integreted model, the authors builds a billingual visual instruction tunning dataset, which mixes recgnition, explanation and caption on images with multi-scale video caption. After training on such dataset, the PAM shows superior performance on classification, OCR, region caption, and video caption tasks.

**Questions:**

1. Does the model lost its instruction following ability after such training?
2. Why is SAM2 frozen during training?
3. Can this framework generalize to other understanding tasks? Does SAM2 representations still outperform other models?

**Ethical Concerns:**

["NO or VERY MINOR ethics concerns only"]

**Final Justification:**

I have carefully considered the paper and the authors' rebuttal. The authors have addressed my concerns. After thorough evaluation, I have decided to raise my recommendation to Borderline Accept (4).

**Limitations:**

Yes

**Quality:**

2

**Strengths And Weaknesses:**

Strengths:
1. The idea of using SAM2 as visual encoder for LLMs to provide better capability on specific understanding tasks is novel.
2. The paper is well written and easy to follow.
3. The experiments shows that PAM can outperform existing MLLMs on various tasks.

Weaknesses
1. Does the model lost its instruction following ability after such training? According to Figure 2 and Section 4, it seems the LLM takes aligned SAM2 outputs and gives the label, discription, and/or caption directly during training. If so, the LLM could lost its instruction following ability and cannot give perception result for text beased prompts (like "the animal behind the door"). Also, it will lost the LLM's ability as fundation model, which highly limits the application of this framework compared to methods like DAM [1].
2. The reason why SAM2 is frozen during the training is not clear. Typically, when applying visual alignment with a vision encoder and LLM, we will (in some stages) finetune the vision encoder for better performance [2,3]. However, in this framework, the SAM2 is completely forzen, and the reason of such conter-intuative design is not well illustrated.
3. Compared to standard pretrained ViTs, representations from SAM2 naturally contains more information on segmentation, therefore is reasonable to have better performance on specific tasks like recognition, explaination, and caption. However, such structure and representation seems not effective on other understanding tasks (e.g., visual question answering), limiting the generalization of this framework.

Overall, although the framework shows superior performance on multiple perception tasks, such gain seems sacrifice the instruction following and generalization ability of LLM, which significantly limits the application of both framework and the trained model compared to similar prior approaches [1].

---

> ### Author Rebuttal · Authors · 2025-07-31
>
> ## **Response to W1 and Q1**
>
> Thanks for your comments. However, we think this concern is based on a misunderstanding of our research. Here are the reasons:
>
> > #### **1. Does the model lost its instruction following ability after such training?**
>
> Our goal was to align the text-only LLM (Qwen2.5) with SAM 2 to create a VLM that excels at powerful region-level visual caption. After training, PAM is endowed with **new visual instruction-following abilities**, including open-set classification, functional explanation, brief and detailed captioning, and streaming video captioning. Therefore, the LLM’s original ability to follow text instructions is **not a key factor in PAM.**
>
> > #### **2. The LLM could lost its instruction following ability and cannot give perception result for text beased prompts.**
>
> Using text-based prompts to identify objects is a feature of general VLMs; the Qwen2.5 LLM, however, never had this capability, as it is unable to process multimodal data. Moreover, we designed PAM to use visual prompts (points, boxes, masks) from SAM 2 because they offer higher precision in complex scenes where describing an object with text can be difficult and ambiguous.
>
> > #### **3. It will lost the LLM's ability as foundation model, which highly limits the application of this framework compared to methods like DAM.**
>
> We think that by integrating two powerful foundation models (SAM 2 and Qwen2.5), we can create a new and robust region-level vision-language foundation model, excelling in fine-grained visual understanding. Our results show this specialized model achieves superior performance on multiple tasks compared to other methods.
>
> For DAM, it is also specialized, focusing primarily on captioning tasks while not supporting others like VQA. Furthermore, DAM is less flexible, as it only accepts mask-based visual inputs. In our experiments, PAM-3B outperforms DAM-3B on multiple benchmarks, demonstrating its effectiveness.
>
> | | LVIS (Img Cls.) | | Ref-L4 (Img Cap.) | | | VideoRefer (Video Cap.) |
> | :--- | :---: | :---: | :---: | :---: | :---: | :---: |
> | | Sem. Sim. | Sem. IoU | ROUGE-L | METEOR | CIDEr | Avg. |
> | DAM-3B | 87.2 | 75.3 | 30.8 | 17.2 | 56.4 | 2.99 |
> | PAM-3B | **88.6** | **78.3** | **31.1** | 17.2 | **59.7** | **3.14** |
>
>
> ## **Response to W2 and Q2: Why is SAM2 frozen?**
>
> Here are the reasons:
>
> 1.	The design of PAM, has two main goals: (1) to generate precise masks based on visual prompts, and (2) to provide rich semantic information for those masks. If it was unfrozen and retrained, the SAM2’s existing segmentation capability could be degraded. While it’s possible to train both SAM2’s segmentation and the LLM’s visual understanding simultaneously, this approach would not only require significantly more computational resources but also introduce a more complex multi-task learning challenge, making the model more difficult to optimize. Therefore, freezing SAM2 is a far more efficient strategy for the overall training process.
>
> 2.	SAM2 inherently contains rich pixel-level information. A key part of our work was to figure out how to best select latent features from SAM2 to efficiently train a region-level VLM. In our ablation studies, we investigated how choosing different latent features impacts model performance, and we analyze the potential reasons in the Methods section. Our experiments demonstrate that it is possible to achieve a high-performing PAM model without retraining SAM2.
>
> 3.	To further validate this, we ran an additional experiment. We trained PAM-1.5B from stage 1.5 checkpoint on 0.5M region-level classification data under two conditions: once with SAM2's parameters frozen, and once with them unfrozen. The results showed that unfreezing SAM2 offered no significant performance boost.
>
> | | **LVIS** | | **PACO** | |
> | :--- | :---: | :---: | :---: | :---: |
> | PAM-1.5B | Sem. Sim. | Sem. IoU | Sem. Sim. | Sem. IoU |
> | -SAM2 unfrozen | **85.9** | 74.2 | 81.9 | **71.9** |
> | -SAM2 frozen | 85.7 | **74.3** | **82.0** | 71.6 |
>
>
> ## **Response to W3 and Q3: Can this framework generalize to other understanding tasks?**
>
> Although PAM was designed to specialize in accurate and detailed region-level descriptions, its architecture inherently supports other understanding tasks (e.g., visual question answering) without requiring any modifications.
>
> To demonstrate this, we collected and trained our model on several open-source, region-level VQA datasets, such as MDVP-Instruct, VCR, Visual7W, and Osprey-724K. The training data were constructed in a format like, "What kind of clothes does the man wear in this \<region\>”
>
> The experimental results are as follows:
>
> | | **Ferret Bench** |  | **VCR** | |
> | :--- | :---: | :---: | :---: | :---: |
> | | Reasoning | Q -> A (%) | QA -> R (%) | Q -> AR (%) |
> | Osprey-7B | 67.8 | - | - | - |
> | Ferret-7B | 67.3 | - | - | - |
> | VILLA-L | - | 78.4 | 82.5 | 65.1 |
> | GPT4RoI-7B | - | 87.4 | **89.6** | 78.6 |
> | **PAM-3B (ours)**| **68.1** | **87.9** | 89.3 | **79.1** |

---

> > ### Comment · Reviewer_H65M · 2025-08-05
> >
> > Thank you for the detailed and thoughtful response. I sincerely appreciate the effort you've put into addressing my concerns. I have no further questions at this time, and I will accordingly raise my score.

---

> > > ### Author Response · Authors · 2025-08-06
> > > **Thanks for your recognition of our rebuttal!**
> > >
> > > Dear Reviewer H65M,
> > >
> > > **Thank you for acknowledging our response and efforts!**
> > >
> > > Regards,
> > >
> > > PAM Authors

---

> ### Author Response · Authors · 2025-08-04
>
> Dear Reviewer H65M,
>
> Many thanks for your time in reviewing our paper and your constructive comments. We have submitted the point-to-point responses on **instruction-following ability**, **why freezing SAM2**, and **generalization**. We would greatly appreciate it if you could let us know whether your concerns have been addressed. We are also happy to provide further clarifications if needed.
>
> Best regards,
>
> Authors of paper #3605 *Perceive Anything: Recognize, Explain, Caption, and Segment Anything in Images and Videos*

---

### Decision · Program_Chairs · 2025-09-17

**Decision:**

Accept (poster)

**Comment:**

The paper initially received 2 Borderline Accepts and 1 Borderline Reject ratings. While reviewers praised the presentation, effectiveness, experiments and interesting ideas, there were concerns on limitations for general visual-language tasks, unclear performance gains and missing experimental comparisons. The rebuttal helped significantly and assuaged most concerns, leading the negative reviewer to upgrade its score to favor acceptance as well, leading to all reviewers favoring acceptance. The AC agrees with the unanimous assessment and believes that the paper has merit, recommending it to be published at NeurIPS. We require that the clarifications, experiments and additional information provided in the rebuttal be incorporated in the camera-ready version.